# Mode coupling bi-stability and spectral broadening in buckled carbon nanotube mechanical resonators

Sharon Rechnitz[1,2], Tal Tabachnik[1,2], Michael Shlafman[1], Shlomo Shlafman[1] & Yuval E. Yaish ●[1]✉

Bi-stable mechanical resonators play a significant role in various applications, such as sensors, memory elements, quantum computing and mechanical parametric amplification. While carbon nanotube based resonators have been widely investigated as promising NEMS devices, a bi-stable carbon nanotube resonator has never been demonstrated. Here, we report a class of carbon nanotube resonators in which the nanotube is buckled upward. We show that a small upward buckling yields record electrical frequency tunability, whereas larger buckling can achieve Euler-Bernoulli bi-stability, the smallest mechanical resonator with two stable configurations to date. We believe that these recently-discovered carbon nanotube devices will open new avenues for realizing nano-sensors, mechanical memory elements and mechanical parametric amplifiers. Furthermore, we present a three-dimensional theoretical analysis revealing significant nonlinear coupling between the in-plane and out-of-plane static and dynamic modes of motion, and a unique three-dimensional Euler-Bernoulli snap-through transition. We utilize this coupling to provide a conclusive explanation for the low quality factor in carbon nanotube resonators at room temperature, key in understanding dissipation mechanisms at the nano scale.

A bi-stable system is the underlying operation principle of modern technology. It is the basic building block for computing, memories, and digital electronics. In MEMS technology[1], bi-stability is achieved by an arch-shaped beam with two symmetric possible configurations: either buckled up (or right) or down (or left). The transition between the two buckled configurations, named Euler–Bernoulli (EB) snap-through (ST) bi-stability[2], is a fascinating example of nonlinear behavior where the compression of the beam decreases its effective spring constant until it vanishes. The transition between the two configurations is controlled by an external force (usually electrostatic) and near this critical transition point, any small force perturbation results in a significant mechanical response, utilized for the realization of ultrasensitive force, acceleration and position sensors. Reducing the dimensions of a MEMS resonator to the nano scale improves its performance[3–6] and enables observation of quantum effects that are not accessible via MEMS devices[7].

Carbon nanotube (CNT) based resonators have been widely investigated for sensing[8,9], signal processing[10], and quantum research[11–15]. However, a bi-stable CNT resonator has never been demonstrated. In this work, we realize the EB snap-through buckling instability in novel suspended CNT resonators and investigate, both theoretically and experimentally, the static and the dynamic behavior of the system, and visualize the complete three-dimensional CNT motion. In fact, we present the first devices in which the CNT is initially buckled upward. This configuration enables unique out-of-plane static motion, which results in a completely different behavior than traditional CNT resonators. The realization of this type of device opens new avenues, some of which we report here: robust bi-stability for long-term endurance of the device, tenfold enhancement over the best electrical frequency tunability reported, and strong mode coupling explaining the anomalous dissipation of CNTs at room temperature.

[1]Andrew and Erna Viterbi Faculty of Electrical and Computer Engineering, Technion, Haifa 3200003, Israel. [2]These authors contributed equally: Sharon Rechnitz, Tal Tabachnik. ✉e-mail: yuvaly@technion.ac.il

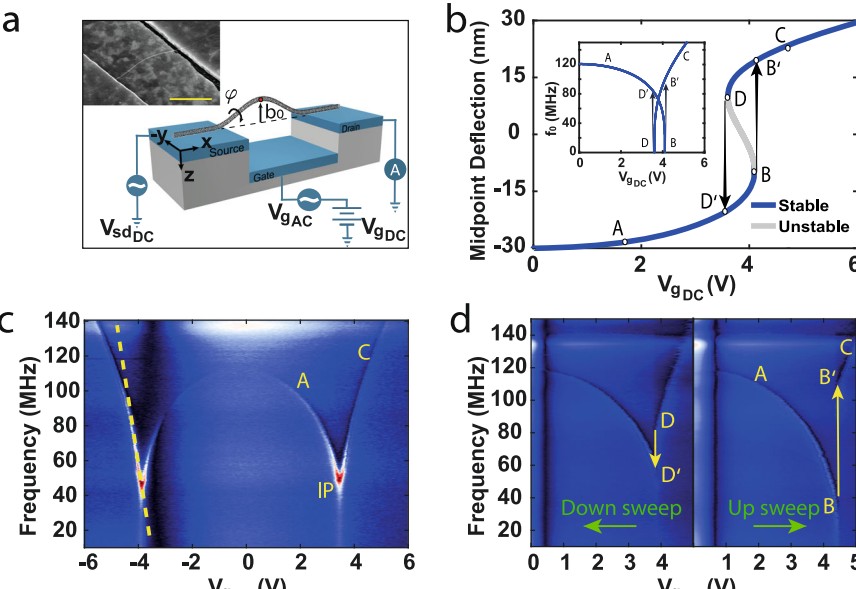

**Fig. 1 | Suspended CNT-based bi-stable NEMS device. a** Schematic layout of the device, coordinates system and experimental setup for the resonance frequency measurement. Note that our coordinates system is such that the positive z direction points downward. Inset: SEM image of a typical device, displaying initial upward buckling. Scale bar−1 μm. **b** Theoretical 2D model for the midpoint static deflection (red dot in a) as a function of the DC gate voltage. Our coordinates were chosen such that negative/positive values represent upward/downward curvature

(respectively). The blue line represents a stable solution while the gray is unstable. Inset is the corresponding theoretical resonance frequency dependence on the DC gate voltage. **c** Resonance frequency measurement of device I, displaying continuous transition via an inflection point (IP), exhibiting $df_0/dV_G$ -100 MHz/V (dashed yellow line). **d** Upward (right) and downward (left) sweeps measurement of device II, exhibiting the snap-through and release phenomena, resulting in a "jump" of nearly 80 MHz in the resonance frequency.

An important characteristic of CNT resonators is the emergence of nonlinear behavior even at a nanometric vibrational amplitude[16,17]. In devices exhibiting EB buckling, the geometric nonlinearity[18,19] becomes more significant since we can observe the transition from compression to stretching. This transition is accompanied by large modulation of the resonance frequencies and strong nonlinear coupling between different modes via the built-in strain. We utilize this induced coupling by the axial strain between the in- and out-of-plane mechanical motions to provide a decisive explanation for the long-standing problem of low-quality factor of CNT resonators at room temperature, in agreement with the fluctuation broadening theory[20].

## Results
### Device structure and experimental setup
Figure 1a presents the geometric structure of our devices. The fabrication is based on the technique developed recently for self-aligned local-gate suspended CNT devices[21]. Briefly, CNTs are grown by chemical vapor deposition (CVD) to create a contact between the source and drain (SD) electrodes, while suspended above a local-gate electrode (Fig. 1a, see Methods section and supplementary information for fabrication details).

All measurements were conducted in a vacuum chamber under the constant pressure of $10^{-5}$ Torr at room temperature. For the resonance frequency measurements, the actuation of the CNT is done electrostatically by the local-gate, and the detection is done by the frequency mixing technique[22], as illustrated in Fig. 1a.

### Resonance frequency measurements
Figure 1c, d present resonance frequency measurements of two devices (devices I and II, respectively), revealing a qualitatively different behavior from previously reported suspended CNT resonators[22,23]. The "classical" suspended CNT resonance frequency increases with the gate voltage due to an increase in tension. In our devices, however, the resonance frequency exhibits a substantial decrease until a noticeable

minimum. The transition from "fall" to "rise" appears either in the form of a sharp dip via an inflection point (IP, Fig. 1c) or by a large "jump" in the resonance frequency (~80 MHz in Fig. 1d). The IP in Fig.1c is characterized by high tunability (marked by the dashed line) with a slope of 100 MHz/V, the highest electrostatic tunability to date of any NEMS/MEMS device[22,24]. A High $df_0/dV_g$ ratio is essential for achieving ultra-sensitive nano-sensors[3], realizing nanomechanical computing[25], and obtaining high-gain parametric amplification and self-oscillation.

Figure 1b presents results from a standard 2D theoretical model that should predict the static response of a buckled beam exhibiting ST phenomenon. Initially, the CNT is buckled upward (see Fig. S4 for SEM images of similar devices). Applying a DC gate voltage attracts the CNT towards the local-gate, generating compression and therefore softening the CNT spring constant, which causes the decrease in the resonance frequency (point A in Fig. 1b–d). If the initial buckling is above a critical instability, an unstable solution is formed (marked in gray, Fig. 1b). Then, the transition between the two stable configurations (in our case, from initial upward buckling to downward buckling) occurs through an abrupt mechanical transition, known as the snap-through Euler–Bernoulli buckling transition[19,26]. This mechanical "jump" causes a change in the spring constant, which translates into a discontinuity in the resonance frequency (B → B' in Fig. 1b, d). After the snap, decreasing the applied voltage, a second critical point is reached (D), resulting in an abrupt transition back to the original geometric configuration, known as the release (D → D'). The ST and release points may occur for different static loads (as is depicted in Fig. 1b), which then translates into a hysteretic response, evidenced in both the DC conductance measurement (see supplementary text) and the resonance frequency measurement (Fig. 1d).

## Discussion
We divide our devices into four different categories (Fig. 2 and S3). In the first category (device III, see Fig. 2a), the CNT is initially buckled downward ($b_0 > 0$), thus exhibiting the common behavior of CNT resonators which has been widely studied[22,24,27].

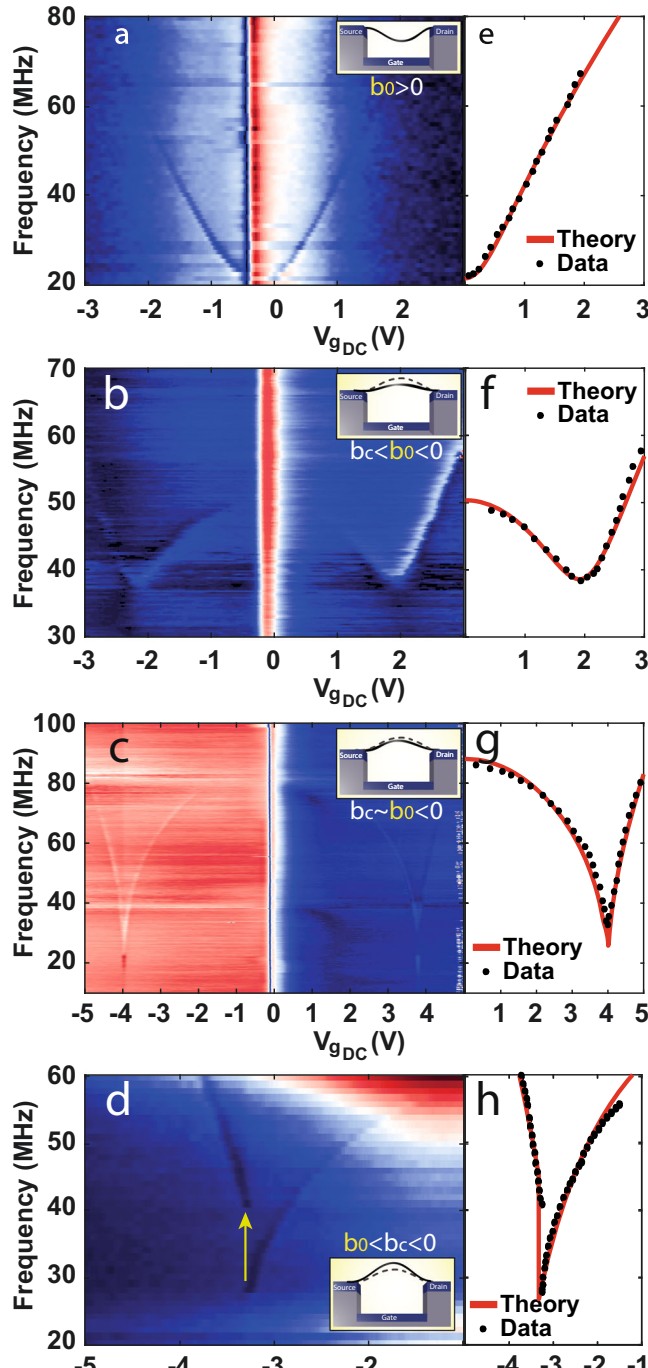

**Fig. 2 | Influence of initial buckling on the dynamic response. a–d** Representative resonance frequency measurements of devices III-VI for the different categories, where **a** $b_0 > 0$, **b** $b_c \ll b_0 < 0$, **c** $b_c \lesssim b_0 < 0$, and **d** $b_0 < b_c < 0$. The insets are schematics of the initial shape of the CNT (solid arch), illustrating the different classes relative to the critical value $b_c$ (dashed arch). **e–h** Modeling the dynamic response of the same devices in (**a–d**), respectively. The solid lines are the theoretical fitting to the experimental data (black dots). The initial CNT heights extracted from the fit are **e** $b_0 = 10$ nm for device III, **f** $b_0 = -6$ nm for device IV, **g** $b_0 = -38$ nm for device V, and **h** $b_0 = -45$ nm for device VI, in agreement with our qualitative interpretation.

The other categories, however, deal with the case of initial upward buckling ($b_0 < 0$), a regime never explored before in CNT resonators. These classes are characterized by the ratio between $b_0$ and the critical initial buckling, $b_c < 0$, for which the ST instability is formed.

In the second group, the initial height is still minor compared to the critical value, meaning $|b_0| \ll |b_c|$. The resonance frequency gate

dependence in this regime is characterized by a mild dip in the resonance frequency measurement (device IV, Fig. 2b). As $|b_0|$ increases, we enter the third category, in which the mild dip sharpens (devices V and I, Figs. 2c and 1c, respectively), but the transition is still continuous. When $|b_0|$ increases above the critical value, we enter the fourth type ($|b_0| > |b_c|$), characterized by a hysteretic static and dynamic response. As a result, a noticeable "jump" in the resonance frequency is observed (devices II and VI in Figs. 1d and 2d, respectively).

Unlike previous CNT bi-stable devices that relied only on pull-in[4], which deteriorates after only several switches[28], our ST transition is robust. The snap-through transition occurs in every DC gate sweep. Hence, during a resonance frequency measurement, it is repeated hundreds of times. For example, we extracted an average ST voltage from the measurement in Fig. 1d of $V_{ST} = 4.39$ V and a standard deviation of $\sqrt{\langle \triangle V_{ST}^2 \rangle} = 0.02$ V. The relative deviation in the snap-through voltage ($\sqrt{\langle \triangle V_{ST}^2 \rangle}/V_{ST} = 0.46\%$) in our CNT resonators is significantly smaller than in previous mentioned bi-stable nanobeams ($V_{ST} \sim 50$ V, $\sqrt{\langle \triangle V_{ST}^2 \rangle}/V_{ST} = >10\%$)[29]. This shows that the ST transition in our CNT resonators is sharp and requires significantly lower power consumption.

While these results are highly encouraging, they leave us with a puzzle. According to the 2D model (Fig. 1b), the ST transition occurs at extremum points, which symbolize zero spring constant and hence zero resonance frequency ($f_0 = 0$). However, looking closer at Figs. 1d, 2d (of devices in the fourth category), we note that the "jump" does not take place at $f = 0$, but at 20–60 MHz above zero. In addition, for the down sweep in Fig. 1d (left), we observe a negative jump, which cannot be explained by the naïve model. To solve this puzzle, we had to realize that the model in Fig. 1b assumes that the CNT movement is constrained to the $xz$-plane (in-plane), where in fact, there is a "hidden" out-of-plane component responsible for our observations.

In order to understand the origin of the different classes of devices and their characteristics, we model our devices as doubly clamped beams[19,26,30]. Since the CNT cross section is circular, we must consider the out-of-plane as well as the in-plane motion[16,31]. The out-of-plane motion and the electrostatic force towards the local-gate impose torque along the CNT axis[30], forming a gate-dependent torsion. Thus, the tube develops a twist during its static motion towards the local-gate. As a result, we formulated a 3D theoretical model in which the Euler–Bernoulli equations of motion are solved considering all three degrees of freedom: in-plane ($z$), out-of-plane ($y$) and twist ($\varphi$) (see supplementary text). We should emphasize the fundamental difference whether the CNT is slacked downward (type I) or buckled upward (types III–V), since only the latter can result in static out-of-plane motion. While many CNT studies took into account out-of-plane dynamics[16] and initial deformation[32], we are the first to consider out-of-plane initial and static deflection in the EB equations, such that significant static out-of-plane movement must evolve. This type of motion is new both for suspended CNTs analysis (traditionally slacked downward) as well as for buckled beams analysis, in which the $y$ (width) and $z$ (thickness) beam dimensions are usually different, restricting the motion to be solely in-plane (width $\gg$ thickness). Our simulations yielded excellent agreement with the experimental results (Figs. 2e–h and 3c, d) and allowed us to visualize the complete and unique CNT motion (Fig. 3e, f).

Regarding the puzzle raised earlier, as to why the "jump" occurs at a finite frequency, the theoretical analysis predicts that the lowest out-of-plane mode ($\omega_{out}$) is always lower than its in-plane counterpart ($\omega_{in}$, see Fig. 3c, d), while the torsional vibrational modes ($\omega_\varphi$) are several orders of magnitude higher ($\omega_\varphi/\omega_{out} \propto L/r$, where $r$ and $L$ are the tube radius and length, respectively). Therefore, the mode that reaches zero frequency and dictates the ST transition is actually the out-of-plane mode. Accordingly, the lowest in-plane mode, which is usually the only mode we can observe in our measurements, will automatically exhibit

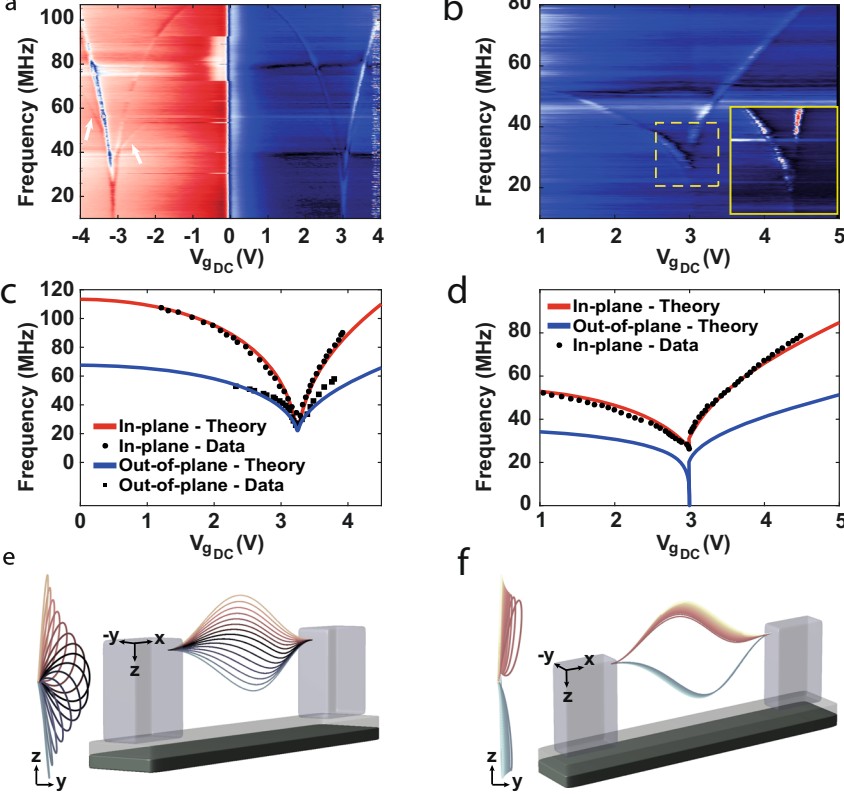

**Fig. 3 | Hybrid in-plane and out-of-plane motion. a** Resonance frequency measurement of device VII, revealing both the out-of-plane (lower, marked by the arrows) and in-plane (second) modes where $b_c \leq b_0 < 0$. **b** Resonance frequency measurement of device VIII, in which $b_0 < b_c < 0$. Only the lowest in-plane mode is detected. Inset is a close-up of the "jump" area marked by the dashed line.

**c, d** Modeling the dynamic response of the same devices as in (**a**) and (**b**), respectively. The solid lines are the theoretical fitting to the experimental data (black dots\squares). **e, f** are the CNT static movement when increasing the gate voltage corresponding to the theoretical fitting in (**c**) and (**d**), respectively.

a "jump" in the resonance frequency even before reaching zero (device VII, Fig. 3b, d). We shall also note that without a twist, a negative jump (as in Fig. 1d left) cannot be accounted for. Occasionally, due to the rotational motion near the inflection point (representing the transition from up to down), we are able to detect evidence for the out-of-plane resonance frequency (marked by the arrows in Fig. 3a). Examining the CNT shape development (Fig. 3e, f) reveals how the in-plane and out-of-plane motions are strongly intertwined.

## Broadening analysis

Lastly, with the experimental data at hand and having a model that reproduces the physical motion of the CNT, we address a long-standing question in the field of CNT resonators. The energy dissipation mechanisms of micro and nano resonators are an important characteristic of the device, commonly represented by their quality factor, $Q$, a quantity indicating how much energy dissipates in one period of oscillation. Unfortunately, the quality factor of CNT resonators at room temperature (~100) is much lower than anticipated (>1000) when considering known dissipation mechanisms[20,22,23,33–35].

An important experimental study showed that spectral broadening due to geometrical symmetry breaking can account for the low $Q$ values at room temperature[23]. However, theoretical simulations suggest another possible mechanism ("fluctuation broadening"), in which thermal fluctuations should cause strong coupling between the in-plane and out-of-plane modes of motion, resulting from the dynamic built-in strain along the CNT[20]. This effect has never been proved experimentally due to the problematic estimation of the built-in strain in standard CNT resonators.

To study the dissipation mechanisms of the system, we calculate the quality factor according to $Q = f_O/\Delta f$, where $\Delta f$ is the full width at

half maximum (FWHM) and $f_O$ is the resonance frequency (example of device IX in Fig. 4a). We compare the $\Delta f$ extracted from the data with a theoretical estimation of $\Delta f$ based on the fluctuation broadening theory (Fig. 4b, see supplementary text for details). In the supplementary text, we also examine the broadening due to symmetry breaking[23,30], which has been previously shown to explain the low $Q$ at room temperature. We show that symmetry breaking cannot account by itself for the spectral broadening evidenced in our measurements (supplementary text).

The broadening due to the in-plane and out-of-plane coupling is estimated according to equation S33 in ref. 20:

$$\sigma_f = \frac{1}{2\pi} \left| \frac{\partial \omega_{ip}}{\partial x_{op}^2} \right| \cdot \sigma_{x_{op}^2} \qquad (1)$$

where $x_{op}$ is the out-of-plane component of the CNT movement with thermal fluctuations given by $\sigma_{x_{op}^2} = \frac{k_B T}{m \omega_{op}^2}$, in which $\omega_{op}$ represents the lowest out-of-plane resonance mode. The peak width is then estimated according to $\Delta f \approx 0.65 \sigma_f$.

As evidenced in Fig. 4b, the fluctuation broadening theory yields remarkable compatibility with the experimental data. This is the most significant experimental evidence to date supporting this theory as the primary cause for low $Q$ in CNT resonators at room temperatures. Moreover, we can show that the other modes' fluctuations contribution to the broadening is negligible and that the lowest out-of-plane mode dominates (see supplementary information). To further examine this theory, we present a similar analysis on device IV at varying temperatures in the supplementary information (Fig. S9), also showcasing excellent agreement between the theoretical prediction and the experimental data.

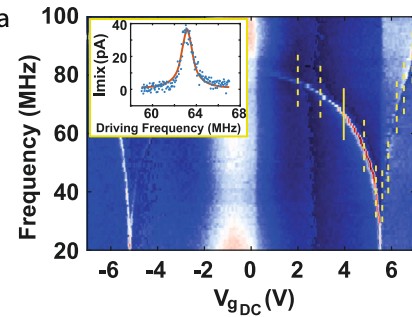

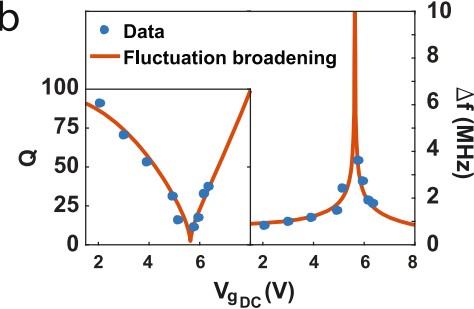

**Fig. 4 | Spectral broadening at room temperature. a** Resonance frequency measurement of device IX in the third category, where $b_c \leq b_0 < 0$. The yellow lines represent the frequency cross sections from which the data in (**b**) were extracted. Inset is an example taken at the yellow solid line. Blue dots are the data, orange solid line is a Lorentzian shape fit. **b** Quality factor (left) and spectral broadening (FWHM, right) extracted from (**a**) vs the static gate load (blue dots). The orange solid line is the theoretical calculation based on Eq. 1. The excellent fit implies that fluctuation broadening is the most significant mechanism to cause the low Q evidenced at room temperature.

To conclude, thanks to a high-quality fabrication process, we were able to identify that there exist several types of carbon nanotube-based resonators and that the main differentiating attribute is the shape of the as-fabricated buckled beam. We found that, unlike the previous common-knowledge, it is possible to fabricate a CNT-based resonator exhibiting Euler–Bernoulli snap-through bi-stability. The technological implication of the resonators classification is that while type A CNT resonators (Fig. 2a) are the standard, the highest electrical frequency tunability is reached in type C (Fig. 2c). In addition, type D devices present a double-well potential CNT resonator, wherein both of the stable states the CNT is fully suspended with no physical contact to any surface. We speculate that this would open the possibility for fabricating integrated mechanical-resonator circuits for advanced computing as well as fast, durable and energy-efficient memory elements. On the theory side, we found that the existence of the snap-through bi-stability is associated with all three degrees of freedom of the CNT. Having a good experiment-model agreement, we also addressed the issue of an anomalously low-quality factor at room temperature and showed that nonlinear mode coupling accounts well for the spectral broadening in our devices, in agreement with the fluctuation broadening theory.

## Methods
### Fabrication
The geometric structure of our devices is presented in Fig. 1a. As a substrate, we used a silicon wafer with a layer of 500 nm silicon oxide on top. All patterning stages were done by standard photolithography or electron beam lithography procedures. We evaporated Cr/Pt 5/35 nm for the source and drain (SD) electrodes, after which we etched the $SiO_2$ using buffer oxide etchant (BOE) to create the local-gate trench (see supplementary text for details). Then, another step of Cr/Pt

5/35 nm evaporation was performed to create the self-aligned local-gate[21]. Finally, we deposited a patterned ferritin catalyst near the SD electrodes for the CNT growth. The CNTs were grown by chemical vapor deposition (CVD) under a constant flow of 0.5/0.5 SLPM $H_2$/$CH_4$ at 900 °C for 20 min, utilizing the fast heating growth technique[36].

### Resonance frequency measurements
The resonance frequency measurements were conducted using the mixing technique[22], presented schematically in Fig. 1a. The actuation of the CNT is done by applying an electrostatic force from the local-gate:

$$F = \frac{1}{2}\frac{\partial C_g}{\partial z}V_g^2 = \frac{1}{2}\frac{\partial C_g}{\partial z}(V_g^{DC} + V_g^{AC}\cos(\omega t))^2 \qquad (2)$$

where $C_g(z)$ is the capacitance between the CNT and the local-gate electrode, $V_g$ is the local-gate voltage, which is a combination of a DC voltage $V_g^{DC}$ superimposed with a driving voltage $V_g^{AC}$ excited at frequency $\omega$.

For the mixing, we apply a second driving voltage between the SD electrodes, $V_{sd}$:

$$V_{sd} = V_{sd}^{AC}\cos(wt + Dwt) \qquad (3)$$

where $V_{sd}^{AC}$ is the driving voltage, and $\Delta\omega$ is the offset frequency (in our experiments, a few kHz). Utilizing the fact that the CNT serves as a mixer, we can actuate it at $\omega$ and detect its resonance frequencies through the current at the offset frequency $\Delta\omega$:

$$I_{mx}^{\Delta\omega} = \frac{1}{2}\frac{\partial G^{DC}}{\partial V_g^{DC}}\left(\frac{1}{C_g}\frac{\partial C_g}{\partial z}V_g^{DC}\delta z + V_g^{AC}\right)V_{sd}^{AC} \qquad (4)$$

where $G^{DC}$ is the DC conductance of the CNT and $\delta z$ is the time-dependent vibrational motion of the CNT.

## Data availability
The raw data used in this study are available in the Zenodo database under accession code https://doi.org/10.5281/zenodo.7051298.

## Code availability
The codes used for the theoretical modeling and analysis in this study are not available as they contain additional valuable insights which have not yet been published. They can be provided upon request from the corresponding author.

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

## Acknowledgements

This study was supported by the ISF (Grant No. 1854/19) and the Russell Berrie Nanotechnology Institute. The work made use of the Micro Nano Fabrication Unit at the Technion. S.R. acknowledges support by the Council for Higher Education and the Russel Berrie scholarships.

## Author contributions

S.R. and T.T. have equal contributions to this work, took part in device fabrication, and performed the experiments and analysis. M.S. developed the fabrication process. S.S. assisted in the theoretical modeling and performed finite element simulations. S.R., T.T., and Y.E.Y. designed the experiment and wrote the manuscript, with input from all authors.

## Competing interests

The authors declare no competing interests.
