## [Peer Review File · Nature Communications]

Title: Mode coupling bi-stability and spectral broadening in buckled carbon nanotube mechanical resonatorsREVIEWER COMMENTS

Reviewer #1 (Remarks to the Author):

Manuscript "Mode coupling, bi-stability, and spectral broadening in buckled carbon nanotube resonators"

By S. Rechnitz et al.

This manuscript describes the experimental and theoretical investigation of nanomechanical carbon nanotube resonators under significant initial buckling. Most interesting, and entirely new to my knowledge, is the case of initial upwards buckling (away from the gate electrode). In this case, the competition between the buckling force and the electrostatic pull enacted by a gate voltage give rise to enormously large resonance frequency tuning and to frequency jumps. The electrostatic tuning of the frequency and the mechanical (static) bistability are by themselves important results that can generate new sensing and logic methods, once the community eventually (hopefully) overcomes the difficulty of mass-fabricating carbon nanotube devices. In addition, however, the authors demonstrate convincingly that the coupling between in-plane and out-of-plane modes is the dominating source for the large damping of carbon nanotubes at room temperature. These results are important and novel and can even be the beginning of a new line of investigation in other groups. The manuscript is written in a very clear manner, free of unnecessary jargon, and it is structured well. The methodology is sound and I have found no reason for technical criticism.

There are only two minor points that I would like to bring to the attention of the authors.

1. On line 136, the authors claim that they are the first to consider out-of-plane initial and static deflection. I would like them to reconsider whether the S-shaped nanotube reported in Phys. Rev. Lett. 109, 025503 (2012), Eichler et al., does not qualify as an example of such static deflection. There, the S shape is modelled in the supplement to obtain good agreement of the resonance frequencies. I would not regard it as a significant reduction of the novelty of the present manuscript if this claim above is somewhat reduced.
2. Ref. [29] is identical with Ref. [9].

Reviewer #2 (Remarks to the Author):

I like the experimental part of this paper, but not the modelling part that indeed is not self-consistent, thus also the related interpretation is questionable. In particular, the authors assume basically a kind of parallel "plate" approximation ($dl=dx$) but impose the presence of the stretching substituting ad hoc F_x with its expression (dl different from dx now); also it is not clear how k is calculated (external force), again assumed only along z ($k=k_z$, whereas it should be perpendicular to the deflected nanotube) and in

any case not constant (how it is varying with the cnt deflection? not clear to me from the text, as the assumed $C_z(z)$); the authors assume the validity of the superposition principle for the displacements that of course is not valid for this problem since nonlinear. This paper is similar to papers I read about 15 years ago ignoring all these aspects, already -at least partially- solved eg see nems papers by Espinosa et al. (that also reported a bi-stable cnt device). Accordingly this paper reports an interesting experiment, a not self-consistent theory, several related best fits. The authors should mitigate these problems and resubmit their paper to a less ambitious journal (or solve them and resubmit here).

Reviewer #3 (Remarks to the Author):

This manuscript investigates a CNT device with a bistable dynamic switching capacity. The device is formed by a CVD-grown doubly-clamped CNT that has an initial upward curve. The resonance frequency response of the CNT device to the driving voltage is measured and a bistable behavior is exhibited in some devices with initial upward curvature. Similar concept of dynamic bistable resonators has been demonstrated before using micro-beams. This work demonstrated this concept using CNTs. Overall, this is an interesting work with potential impacts on the development of nanotube nano devices. However, a general criticism is that the manuscript (including SI) does not provide sufficient data and information that makes the assessment of this work harder. Below are some concerns that the authors should consider to address in the revised manuscript.

(1) The entire manuscript is based on the assumption that the tested nanotube is an initially buckled tube. This is not completely clear based on its growth method and is more like a speculation.

Alternatively, the initial tube configuration could be just a slack beam which is free of internal axial force. More convincing evidence is needed to back the authors' assumption/speculation.

(2) The authors should provide more information about the CNTs that were grown and tested in this work. For examples, are they single- or multi-walled? What are their inner and outer diameters?

(3) For the static (and also dynamic) modeling, no parameters about the nanotubes (such as length, diameter, Young's modulus, initial mid-point elevation b_0 , etc) as well as the trench dimensions were given. It is not clear what are the parameters used to obtain the theoretical curve shown in Fig. 1b.

(4) What is the initial mid-point elevation of the tube (b_0) for the device simulated in Fig. 1b and what is the corresponding critical buckling value (b_c)? The max deflection of the tube's mid-point is ~ 10 nm. If the initial b_0 is ~ 30 - 40 nm, does the nanotube remain in an upward configuration during the snap-through transition? If this is true, it is not clear there would be any compression to tension transition inside the tube during the switching process.

(5) The accompanying out of plane resonance is interesting. Authors should comment how the out-of-plane resonance would affect the switching of the bistable states. Is there any advantage of suppressing such out of plane resonance in terms of obtaining better bistable dynamic switching properties and how?

(6) Authors should clarify how many nanotube devices were actually tested in each listed device categories.

21/04/2022

Dear Reviewers,

We carefully read your reviews and would like to thank you for the professional and overall positive feedback.

Here is our point-by-point response. The changes we have made within the manuscript were marked by yellow.

Reviewer #1

This manuscript describes the experimental and theoretical investigation of nanomechanical carbon nanotube resonators under significant initial buckling. Most interesting, and entirely new to my knowledge, is the case of initial upwards buckling (away from the gate electrode). In this case, the competition between the buckling force and the electrostatic pull enacted by a gate voltage give rise to enormously large resonance frequency tuning and to frequency jumps. The electrostatic tuning of the frequency and the mechanical (static) bistability are by themselves important results that can generate new sensing and logic methods, once the community eventually (hopefully) overcomes the difficulty of mass-fabricating carbon nanotube devices. In addition, however, the authors demonstrate convincingly that the coupling between in-plane and out-of-plane modes is the dominating source for the large damping of carbon nanotubes at room temperature. These results are important and novel and can even be the beginning of a new line of investigation in other groups. The manuscript is written in a very clear manner, free of unnecessary jargon, and it is structured well. The methodology is sound and I have found no reason for technical criticism.

There are only two minor points that I would like to bring to the attention of the authors.

1.1. On line 136, the authors claim that they are the first to consider out-of-plane initial and static deflection. I would like them to reconsider whether the S-shaped nanotube reported in Phys. Rev. Lett. 109, 025503 (2012), Eichler et al., does not qualify as an example of such static deflection. There, the S shape is modelled in the supplement to obtain good agreement of the resonance frequencies. I would not regard it as a significant reduction of the novelty of the present manuscript if this claim above is somewhat reduced.

Thank you for your comment, and we apologize if you felt that our claim was exaggerated. You are correct that the S-shaped tube has initial out-of-plane deflection, and it is very reasonable to assume that the authors' ANSYS modelling takes out-of-plane statics into account. However, their Euler-Bernoulli based analytic analysis did not take the out-of-plane motion into account, nor do they discuss the out-of-plane static motion in their text. This makes sense as in the S-shapes case, applying DC gate voltage should only reduce out-of-plane deflection, as opposed to our case, in which substantial out-of-plane deflection evolves with the gate voltage. Nevertheless, we have rephrased the sentence to acknowledge the referred work, and we hope that you find this rephrasing more suitable:

"While many CNT studies took into account out-of-plane dynamics¹⁶ and initial deformation³², we are the first to consider out-of-plane initial and static deflection in the EB equations, such that significant static out-of-plane movement must evolve."

1.2. Ref. [29] is identical with Ref. [9].

Thank you for pointing this out, the redundancy was fixed in the revised manuscript.

Reviewer #2

I like the experimental part of this paper, but not the modelling part that indeed is not self-consistent, thus also the related interpretation is questionable.

2.1. In particular, the authors assume basically a kind of parallel "plate" approximation ($dl=dx$) but impose the presence of the stretching substituting ad hoc F_x with its expression (dl different from dx now);

You are correct that F_x results from the self-induced tension along the tube, which essentially means that dl is different from dx . However, the shallow arch approximation ($dl \sim dx$ when $b_0 \ll L$ and $d \ll L$), is the standard method in literature for the modelling of arch-shaped beams when using the EB equations (*Krylov et al., Continuum Mech. Thermodyn. (2010)* / *Younis et al., Journal of Microelectromechanical Systems (2010)* / *Landau & Lifshitz, Theory of Elasticity (1986)*). In our devices, the "beam thickness" is the CNT diameter typically $d \sim 1-4\text{nm}$, the initial maximum deflection is typically $b_0 \sim 30-50\text{nm}$, and the tube length, L , ranges from 600nm to 1500nm ; hence, both $b_0 \ll L$ and $d \ll L$. Eliminating the twist reduces the equations based on the force moment (equations S3) to the familiar EB beam equations (similar to the formulation in *Landau & Lifshitz, Theory of Elasticity (1986)*).

2.2. also it is not clear how k is calculated (external force), again assumed only along z ($k=kz$, whereas it should be perpendicular to the deflected nanotube) and in any case not constant (how it is varying with the cnt deflection? not clear to me from the text, as the assumed $C_z(z)$);

The force is indeed assumed to be only along z , as the external force is exerted by the local gate perpendicular to the gate plane, which is along the z axis. However, we do NOT assume constant force, since, as you've mentioned, the capacitance is a function of the tube's deflection. We apologize if this was not clear enough in the text. Thanks to your comment, we have added the following clarification on how the force is calculated to the SI:

κ_z is the electrostatic force exerted by the local-gate, calculated according to: $\kappa_z = \frac{1}{2} \frac{\partial C_g}{\partial z} (V_{gDC} + V_{gAC})^2$. We assume $V_{gAC} \ll V_{gDC}$ so $(V_{gDC} + V_{gAC})^2 \approx V_{gDC}^2 + 2V_{gDC}V_{gAC}$ and use the first term as the static force and the second term for the harmonic actuation. The capacitance is calculated according to a "wire parallel to plane" approximation, which, under the assumption $\frac{(g_0+w)^2}{r^2} \gg 1$, can be approximated as

$$C_g(z) = \frac{2\pi\epsilon_0}{\ln\left(\frac{2(g_0+z)}{r}\right)} \Rightarrow \frac{\partial C_g}{\partial z}(z) = \frac{\pi\epsilon_0}{(g_0+z) \left(\ln\left(\frac{2(g_0+z)}{r}\right)\right)^2}.$$

We hope you find this calculation of the force adequate.

2.3. the authors assume the validity of the superposition principle for the displacements that of course is not valid for this problem since nonlinear.

The superposition principle separating the beam motion to initial + static + dynamic is vastly used to solve the arch-shaped resonators as well as CNT resonators. For example: *Krylov & Dick, Continuum Mech. Thermodyn. 22:445–468 (2010)*; *Najar et al., Int. J. Mech. Sci. 178 105624 (2020)*; *Ouakad & Younis, Journal of Sound and Vibration 330, 3182-3195 (2011)*. The nonlinear terms in the EB

equations arise from the built-in tension along the beam. Since the beam deflection is much smaller than its length, it is possible to approximate the tension as integral of the derivative square. Then, we keep all the relevant terms and solve the coupled nonlinear equations. To the best of our knowledge, superposition of in-plane and out-of-plane motions has also been used before in nonlinear problems, and even for carbon nanotubes, see for example *Conley et al., Nano Letters (2008)* and *Ouakad and Younis, Journal of Sound and Vibration 330, 3182-3195 (2011)*.

2.4. *This paper is similar to papers I read about 15 years ago ignoring all these aspects, already -at least partially- solved eg see nems papers by Espinosa et al. (that also reported a bi-stable cnt device).*

We carefully checked several papers by Espinosa et al. regarding CNTs and NEMS. We believe that your comment and critique are based on the theoretical framework presented in Espinosa et al. studies, especially in the following three articles:

1. Analysis of Doubly Clamped Nanotube Devices in the Finite Deformation Regime, N. Pugno, C. H. Ke, H. D. Espinosa, *Journal of Applied Mechanics*, **72**, 445 (2005).
2. Numerical Analysis of Nanotube-Based NEMS Devices—Part I: Electrostatic Charge Distribution on Multiwalled Nanotubes, Changhong Ke, Horacio D. Espinosa, *Journal of Applied Mechanics*, **72**, 721 (2005).
3. Numerical Analysis of Nanotube Based NEMS Devices — Part II: Role of Finite Kinematics, Stretching and Charge Concentrations, Changhong Ke, Horacio D. Espinosa, Nicola Pugno, *Journal of Applied Mechanics*, **72**, 726 (2005).

The main point in these articles is the following:

In the Euler-Bernoulli beam equation for a suspended CNT exerted an electric force toward the gate electrode one should introduce a stretching term, a finite kinematics term, and an electrostatic charge distribution term. Overall, the EB equation should look as follows:

$$E \cdot I \cdot \frac{d^2}{dx^2} \frac{\frac{d^2 w(x)}{dx^2}}{\left(1 + \left(\frac{dw(x)}{dx}\right)^2\right)^{3/2}} - \frac{E \cdot A}{2L} \int_0^L \left(\frac{dw(x)}{dx}\right)^2 dx \cdot \frac{\frac{d^2 w(x)}{dx^2}}{\left(1 + \left(\frac{dw(x)}{dx}\right)^2\right)^{3/2}} = \frac{\pi \epsilon_0}{\left(\log(2(g_0 - w(x)) / r)\right)^2} \frac{1}{g_0 - w(x)} \cdot V_g^2 \cdot \left(1 + \left(\frac{dw(x)}{dx}\right)^2\right)^{1/2}$$

All the letters we used are the same as in our manuscript. Shortly, E is the tube Young's modulus, I is moment of inertia, w is the vertical static displacement, x is the horizontal axis, g₀ is the vertical distance between a straight CNT and the gate electrode, L is the tube length, r is the radius, and V_g is the applied gate voltage. The red terms represent the finite kinematics terms. The green term is the tube stretching term. The blue term is the electrostatic charge distribution term. The above-mentioned papers refer to three possible scenarios:

1. Small deformation, in which dw/dx << 1, and the familiar EB beam equation is recovered:

$$E \cdot I \cdot \frac{d^4 w(x)}{dx^4} = \frac{\pi \epsilon_0}{\left(\log(2(g_0 - w(x)) / r)\right)^2} \frac{1}{g_0 - w(x)} \cdot V_g^2$$

2. Moderate deformation, in which $(dw/dx)^2 \ll 1$, then the EB eq. is the following:

$$E \cdot I \cdot \frac{d^4 w(x)}{dx^4} - \frac{E \cdot A}{2L} \int_0^L \left(\frac{dw(x)}{dx} \right)^2 dx \cdot \frac{d^2 w(x)}{dx^2} = \frac{\pi \varepsilon_0}{(\log(2(g_0 - w(x))/r))^2} \frac{1}{g_0 - w(x)} \cdot V_g^2.$$

3. Finite kinematics, where the full EB beam equation should be solved.

One possible way to estimate the importance of each term is by rewriting the EB equation in dimensionless units. Let us introduce the new dimensionless variables:

$$x = \tilde{x} \cdot L \quad ; \quad w = \tilde{w} \cdot g_0 \quad ; \quad r = \tilde{r} \cdot g_0$$

After plugging these definitions into the full EB equation, we find:

$$\frac{E \cdot I \cdot g_0}{L^4} \cdot \frac{d^2}{d\tilde{x}^2} \frac{\frac{d^2 \tilde{w}(\tilde{x})}{d\tilde{x}^2}}{\left(1 + \left(\frac{g_0}{L} \right)^2 \left(\frac{d\tilde{w}(\tilde{x})}{d\tilde{x}} \right)^2 \right)^{3/2}} - \frac{E \cdot A \cdot g_0^3}{2L^4} \int_0^1 \left(\frac{d\tilde{w}(\tilde{x})}{d\tilde{x}} \right)^2 d\tilde{x} \cdot \frac{\frac{d^2 \tilde{w}(\tilde{x})}{d\tilde{x}^2}}{\left(1 + \left(\frac{g_0}{L} \right)^2 \left(\frac{d\tilde{w}(\tilde{x})}{d\tilde{x}} \right)^2 \right)^{3/2}} = \frac{\pi \varepsilon_0}{\left(\log \left(\frac{2(1 - \tilde{w}(\tilde{x}))}{\tilde{r}} \right) \right)} \frac{1}{g_0(1 - \tilde{w}(\tilde{x}))} \cdot V_g^2 \cdot \left(1 + \left(\frac{g_0}{L} \right)^2 \left(\frac{d\tilde{w}(\tilde{x})}{d\tilde{x}} \right)^2 \right)^{1/2}$$

And after dividing the last Eq. by $E I g_0 / L^4$ we obtain the dimensionless EB beam equation:

$$\frac{d^2}{d\tilde{x}^2} \frac{\frac{d^2 \tilde{w}(\tilde{x})}{d\tilde{x}^2}}{\left(1 + \left(\frac{g_0}{L} \right)^2 \left(\frac{d\tilde{w}(\tilde{x})}{d\tilde{x}} \right)^2 \right)^{3/2}} - \frac{A \cdot g_0^2}{2I} \int_0^1 \left(\frac{d\tilde{w}(\tilde{x})}{d\tilde{x}} \right)^2 d\tilde{x} \cdot \frac{\frac{d^2 \tilde{w}(\tilde{x})}{d\tilde{x}^2}}{\left(1 + \left(\frac{g_0}{L} \right)^2 \left(\frac{d\tilde{w}(\tilde{x})}{d\tilde{x}} \right)^2 \right)^{3/2}} = \frac{\pi \varepsilon_0}{\left(\log \left(\frac{2(1 - \tilde{w}(\tilde{x}))}{\tilde{r}} \right) \right)} \frac{1}{(1 - \tilde{w}(\tilde{x}))} \cdot V_g^2 \cdot \frac{L^4}{E \cdot I \cdot g_0^2} \left(1 + \left(\frac{g_0}{L} \right)^2 \left(\frac{d\tilde{w}(\tilde{x})}{d\tilde{x}} \right)^2 \right)^{1/2}$$

The pre-factor in front of the stretching term can be written as follows: $\alpha = \frac{A \cdot g_0^2}{2I} \approx \frac{2\pi r t \cdot g_0^2}{2\pi r^3 t} = \left(\frac{g_0}{r} \right)^2$

where t is the mechanical interlayer spacing ~ 0.066 nm. This term is quite big in our experimental study. r is a few nm and $g_0 \sim 150$ nm, thus overall $\alpha \approx 10^4$. In non dimensionless units the maximum value of dw/dx will be of the order of g_0/L for a doubly clamped beam with a deflection less than $g_0/3$ (before pull-in). Therefore, in dimensionless units $\frac{d\tilde{w}(\tilde{x})}{d\tilde{x}}$ is of the order of 1. In our experiment

$g_0/L \sim 1/7 - 1/10$, thus the terms $\left(1 + \left(\frac{g_0}{L} \right)^2 \left(\frac{d\tilde{w}(\tilde{x})}{d\tilde{x}} \right)^2 \right) \approx 1 + \left(\frac{1}{50} \text{ or } \frac{1}{100} \right) \approx 1$, and we can safely ignore

0.02 or 0.01 with respect to 1. However, in the stretching term, the small term $\left(\frac{d\tilde{w}(\tilde{x})}{d\tilde{x}} \right)^2$ is of order 1

and it is multiplied by $\alpha \approx 10^4$. In the force expression, the fact that the force is not constant is mainly expressed by this term: $\frac{1}{(1-\tilde{w}(\tilde{x}))}$. Again, the maximum value of $\tilde{w}(\tilde{x})$ is of the order 1/3. Therefore,

both the stretching term and the non-constant force term are important and cannot be ignored. It is correct that for the cases of pull in experiments with single or double clamped beams, it could be that the derivative of dw/dx near the collapsed segment is bigger than g_0/L , but in our experiment, we are far from these extreme cases.

Indeed, in the articles of Espinosa et al. they compare the behavior of the CNT under different conditions. Especially they present the CNT shape for two scenarios: for small deflections (which omits both the stretching and kinematics terms) and for full kinematics (includes the charge distribution term). Below is Fig. 5 from article #3 in the list above:

Fig. 5 Elastic line for fixed-fixed nanotube at $V=5$ V. The solid line is for finite kinematics, the dotted line assumes small deformations.

This result agrees very well with our previous analysis. The stretching term is extremely important in modelling the beam shape under applied vertical force. Without it, the restoring force of the tube is much smaller and consequently the anticipated tube deflection seems much bigger than it actually is. However, in their papers they did not address the question if the full kinematics and the charge distribution terms are necessary, or whether it is sufficient to describe the tube deflections and shape with the stretching term alone (moderate case). In order to adequately address this issue, we decided to solve the EB beam equation introducing all three cases using finite element model (FEM). This way we can point out which term is the most relevant for our experiment.

The FEM we used is the same as described in our theoretical paper (Rechnitz, S. *et al.* Theoretical modeling of arch shaped carbon nanotube resonators exhibiting Euler-Bernoulli snap-through bi-stability (2022) <http://arxiv.org/abs/2203.02738>). The tube length is $1\mu\text{m}$ and the tube radius is 1.1 nm. Below are the obtained results for different external gate voltages. Fig. R1 presents the tube shape for $V_g=1.5\text{V}$.

Fig. R1. CNT shape for different EB approximations.

In the left figure we can easily see how the small deformation case evaluates much bigger displacement compared to the moderate or finite kinematics cases, similar to Espinosa et al. results. In the right image, we zoom only on the two last cases, moderate and finite kinematics. We can see that the two cases predict the same CNT shape. The maximum displacement in the right image is much smaller than g_0 , therefore, this result is expected based on the analysis we discussed before. In order to check the range of validity of the moderate approximation, we calculate the CNT shape for increasing values of V_g . Fig. R2 depicts the results of this calculation.

Fig. R2. CNT shape for different EB approximations and gate voltages.

One can see that only for maximal CNT displacements which are comparable to g_0 a noticeable (although still small) difference between the moderate approximation to the full kinematics model is observed. Since in our experiment the maximal CNT displacement is approximately 50 nm, the above analysis confirms that taking the EB equation with the moderate approximation is justified. If we increase the gate by a small amount above 40V, a larger difference between the two models appears, suggesting that the moderate approximation fails to accurately describe the CNT deflection. Fig. R3 presents the resulted CNT's shape for several gate voltages above 40V.

Fig. R3. CNT shape for different EB approximations and gate voltages above 40V.

Accordingly this paper reports an interesting experiment, a not self-consistent theory, several related best fits. The authors should mitigate these problems and resubmit their paper to a less ambitious journal (or solve them and resubmit here).

We hope that after reading our reply, which addresses each and every point that you have raised you will find our manuscript suitable for publication in Nature Communications. We will be happy to explain any further issue which remains unclear.

Reviewer #3

This manuscript investigates a CNT device with a bistable dynamic switching capacity. The device is formed by a CVD-grown doubly-clamped CNT that has an initial upward curve. The resonance frequency response of the CNT device to the driving voltage is measured and a bistable behavior is exhibited in some devices with initial upward curvature. Similar concept of dynamic bistable resonators has been demonstrated before using micro-beams. This work demonstrated this concept using CNTs. Overall, this is an interesting work with potential impacts on the development of nanotube nano devices. However, a general criticism is that the manuscript (including SI) does not provide sufficient data and information that makes the assessment of this work harder. Below are some concerns that the authors should consider to address in the revised manuscript.

(3.1) The entire manuscript is based on the assumption that the tested nanotube is an initially buckled tube. This is not completely clear based on its growth method and is more like a speculation. Alternatively, the initial tube configuration could be just a slack beam which is free of internal axial force. More convincing evidence is needed to back the authors' assumption/speculation.

We provide several side-view SEM images in the supplementary text which clearly show an initial upward curvature, and we even present a quantitative comparison between our predicted initial

elevation vs. the SEM data with excellent agreement. Unfortunately, we could not perform such measurements for all of our devices, since high resolution SEM imaging can deteriorate the devices, but we have never come across a SEM image of initial curvature qualitatively different than what is anticipated from the resonance measurements. We cannot think of better evidence to support our claim, and according to our knowledge, it is quite common to rely on SEM imaging to determine the initial shape of the tube, just as in the article referred to by Reviewer #1 - *Phys. Rev. Lett.* 109, 025503 (2012), *Eichler et al.* We added a clearer reference to the SI SEM data (Fig. S4) in the main text (line 84). Below are the relevant SEM images (Fig. R4).

If we understand you correctly, you are concerned that upward initial curvature does not necessarily imply initial compressed axial strain, as in our model. Let us explain how the initial compression can be evidenced from the data. In typical resonance frequency measurements (in both literature as well as in our type I devices), the parabolic-shaped resonance curve minimum is shifted from $V_{gDC}=0$. This is explained by the fact that the flat-band voltage is not zero (due to charge transfer, see *Meerwaldt et al., Carbon nanotubes: Nonlinear high-Q resonators with strong coupling to single-electron tunneling (2012)* and *Eichler et al., Nat. Nano. (2011)*), and hence an electrostatic attractive force is exerted upon the CNT by the local gate, even for $V_{gDC}=0$. In our types II-IV devices, we also observe a shift of the maximum frequency from $V_{gDC}=0$, which also implies the existence of an electrostatic force at $V_{gDC}=0$, see example below (Fig. R5). Hence, the CNT cannot remain in an upward configuration unless there is an opposing compressed axial force, forcing the CNT to be buckled upward. Furthermore, we shall point out that taking a zero initial tension cannot achieve good fit to the experimental data, and hence must be inserted to the equations.

Fig. R4. (same as Fig. S4) (a,d) SEM images of initially buckled upward CNTs (devices X and V, respectively), retrieved at a 75° and 70° (respectively) angle from the perpendicular to the surface. (b,e) Resonance frequency measurements of the same devices as in (a) and (d), respectively, typical of the third category (obtained prior to the SEM measurements). Measurement (e) is the same as Fig. 2c. (c,f) Theoretical fitting (red line) to the data (black dots) extracted from the resonance measurement in (b,e), respectively.

Fig. R5. (left) Resonance frequency measurement of Device I (same as in Fig. 1c, showcasing that the maximum frequency at upward-buckled configuration, marked by the yellow line, is obtained for $V_{gDC} \neq 0$ ($V_{gDC} = 0$ is indicated by the grey line)). (right) Zoom-in on the same measurement near $V_{gDC} = 0$. Orange dots are discrete data sampled from the resonance measurement to which the thin grey line was fitted. The maximum is obtained for $V_{gDC} = -0.25V$.

(3.2) The authors should provide more information about the CNTs that were grown and tested in this work. For examples, are they single- or multi-walled? What are their inner and outer diameters?

The CNTs grown and tested in this work are quite diverse – most are single-walled while a few are multi-walled. In order to determine which type, we relied on the diameter measured in AFM, the device transfer characteristics, and micro-RAMAN measurements. For example, Fig. R6 presents micro-Raman spectra of two CNTs listed in the table below (devices I and III). According to the literatures (for example: Puech et al., Journal of RAMAN Spectroscopy, **38**: 714 – 720 (2007)), the two spectra near the G and 2D bands reflect data from single-wall CNTs.

Fig. R6. Raman spectra of devices I (top) and III (bottom).

Once the number of walls was determined, we took this into account in our modelling, specifically in the bending rigidity calculation. Following is a table detailing the physical parameters of each device. This table was added to the SI.

Device	Diameter [nm]	Length [μm]	Number of walls
I	1.9	0.7	1
II	3.5	1.2	2
III	1.8	1.34	1
IV	2.25	0.86	1
V	2	1	1
VI	2.6	0.78	1
VII	3	1.2	2
VIII	1.7	0.9	1
IX	2.5	0.89	1
X	4	1.3	3

(3.3) For the static (and also dynamic) modeling, no parameters about the nanotubes (such as length, diameter, Young's modulus, initial mid-point elevation b_0 , etc) as well as the trench dimensions were given. It is not clear what are the parameters used to obtain the theoretical curve shown in Fig. 1b.

The physical parameters for the modelling of each device were taken according to the specific device parameters. Length, diameter and trench depth were extracted from an AFM image of the device. Thanks to your inquiry, we added a clarification for this in the SI under "influence of initial buckling". The initial CNT shape (i.e. b_0) was a fitting parameter. Only after resonance measurement and fitting, we could compare the fitted initial height vs the b_0 extracted from SEM, which was done for a few devices (not all) to determine the accuracy of the model, as presented in Fig. S4. Specifically, for the device in Fig. 1b,d,e, the physical parameters are: $d=3.5\text{nm}$, $L=1.2\mu\text{m}$, $g_0=154\text{nm}$. However, we shall note that the curve in Fig. 1b is only a naïve 2D model and therefore we do NOT consider its prediction to be correct (as supposed to the model that produces the fitting in Figs. 2 and 3). We believe that the answer to your next question will clarify the confusion.

(3.4) What is the initial mid-point elevation of the tube (b_0) for the device simulated in Fig. 1b and what is the corresponding critical buckling value (b_c)? The max deflection of the tube's mid-point is $\sim 10\text{ nm}$. If the initial b_0 is $\sim 30\text{-}40\text{ nm}$, does the nanotube remain in an upward configuration during the snap-through transition? If this is true, it is not clear there would be any compression to tension transition inside the tube during the switching process.

We apologize for the confusion that arises from this figure. Fig. 1b is based on the naïve 2D model and therefore cannot represent the actual device; it was only meant to help the reader understand the intuitive interpretation discussed in the text, and demonstrate the problem of ST jump from zero frequency. Hence, we chose random parameters in this simulation with very small initial buckling of 4nm , so the CNT does "jump" from upward to downward buckling. Thanks to your comment, we understand why this figure is misleading, so we have replaced it with a simulation of more realistic initial conditions ($b_0=30\text{nm}$). In addition, we changed the y axis to represent the absolute z deflection, as opposed to the relative deflection from the initial state, such that a transition from negative to

positive values represents the transition from upward to downward curvatures. We hope that you find the new figure clearer. Thank you for your comment.

(3.5) The accompanying out of plane resonance is interesting. Authors should comment how the out-of-plane resonance would affect the switching of the bistable states. Is there any advantage of suppressing such out of plane resonance in terms of obtaining better bistable dynamic switching properties and how?

This is a very interesting, important, and not quite trivial question. Indeed, when the out-of-plane and in-plane modes are similar, the transition from upward to downward buckling is continuous. In order to achieve bi-stability, the modes must be farther apart, since the out-of-plane mode should reach zero while the in-plane mode can remain high. The way to control the system mechanics is by the initial conditions, however it is not trivial as bi-stability is affected by the in-plane configuration, out-of-plane configuration and the initial strain. You can refer to our theoretical paper for more details - <https://arxiv.org/abs/2203.02738>.

(3.6) Authors should clarify how many nanotube devices were actually tested in each listed device categories.

Thanks to your inquiry, we have added the number of devices in each category from which the data in this study was extracted to the supplementary text:

1st category (downward slack) – 12 devices

2nd category (small initial upward curvature) – 8 devices

3rd category (more substantial initial upward curvature) – 17 devices

4th category (devices exhibiting ST bi-stability) – 14 devices

We would like to thank all the reviewers again for their feedback. We hope that you find the above point-by-point response and attached revision satisfactory.

Sincerely,

Mrs. Sharon Rechnitz
Dr. Tal Tabachnik
Dr. Michael Shlafman
Dr. Shlomo Shlafman
Prof. Yuval Yaish

REVIEWER COMMENTS

Reviewer #2 (Remarks to the Author):

Thank you for your review, I am not convinced about the answers often reporting references rather than replies/justifications, the most critical example is the superposition principle: the authors mention papers where it is used for solving nonlinear problems whereas for me it assumes linearity by definition, thus if you apply it in a nonlinear system this is wrong, more or less depending on the level of nonlinearity, accordingly the authors should at least evaluate the error in applying it and check if it is acceptable or not.

Eg from https://en.wikipedia.org/wiki/Superposition_principle

The superposition principle,[1] also known as superposition property, states that, for all linear systems, the net response caused by two or more stimuli is the sum of the responses that would have been caused by each stimulus individually. So that if input A produces response X and input B produces response Y then input (A + B) produces response (X + Y).

...

In most realistic physical situations, the equation governing the wave is only approximately linear. In these situations, the superposition principle only approximately holds. As a rule, the accuracy of the approximation tends to improve as the amplitude of the wave gets smaller. For examples of phenomena that arise when the superposition principle does not exactly hold, see the articles nonlinear optics and nonlinear acoustics.

Reviewer #3 (Remarks to the Author):

The authors have addressed my concerns adequately. The manuscript looks ready for publication.

05/07/2022

Dear Editor,

We would like to thank you for inviting a revision to our manuscript. Following is our detailed response to Reviewer #2's concern.

We hope you find this explanation sufficient and approve our manuscript for publication.

Reviewer #2

Thank you for your review, I am not convinced about the answers often reporting references rather than replies/justifications, the most critical example is the superposition principle: the authors mention papers where it is used for solving nonlinear problems whereas for me it assumes linearity by definition, thus if you apply it in a nonlinear system this is wrong, more or less depending on the level of nonlinearity, accordingly the authors should at least evaluate the error in applying it and check if it is acceptable or not.

Eg from https://en.wikipedia.org/wiki/Superposition_principle

The superposition principle, [1] also known as superposition property, states that, for all linear systems, the net response caused by two or more stimuli is the sum of the responses that would have been caused by each stimulus individually. So that if input A produces response X and input B produces response Y then input (A + B) produces response (X + Y).

...

In most realistic physical situations, the equation governing the wave is only approximately linear. In these situations, the superposition principle only approximately holds. As a rule, the accuracy of the approximation tends to improve as the amplitude of the wave gets smaller. For examples of phenomena that arise when the superposition principle does not exactly hold, see the articles nonlinear optics and nonlinear acoustics.

Thank you for your feedback, and we understand your concern; our original response should have been more detailed.

Let us first explain the origin of the non-linearity in the Euler-Bernoulli (EB) beam equation, and to clarify which terms are still linear.

Let's assume that we have a CNT along the x direction and its movements can be either in-plane (x-z plane) or out-of-plane (x-y plane). Figure 1 depicts the geometry and notation of our problem.

Figure 1. Schematic of a typical bi-stable CNT device with initial upward buckling and our coordinates system.

For our purpose, it is sufficient to limit ourselves to the static deflection (the conclusions for the dynamic case are the same). Let's assume in the beginning that the motion is restricted to the x-z plane and note the static deflection along the z direction by $w(x)$. Assuming initial compression ($-T_0 < 0$) and induced tension which arises from the shape of the beam, $T_{ind}(w(x))$, we end up with the following EB equation:

$$EI \frac{d^4}{dx^4} w(x) + T_0 \frac{d^2}{dx^2} w(x) - T_{ind} \frac{d^2}{dx^2} w(x) = F_{ext} \quad (1)$$

where E is the tube Young's modulus, I is the moment of inertia, and F_{ext} is the external static force per unit length. The non-linearity of this equation arises from two terms. The first is expressed by T_{ind} and the second is related to the external force.

Let's examine first T_{ind} . When the tube is not straight the deflection $w(x)$ makes the overall length (L) longer than its relaxed length, L_0 . Therefore, the induced tension will be given by: $T_{ind} = EA \frac{\Delta L}{L_0} = EA \frac{(L-L_0)}{L_0}$, where A is the tube's cross section. How shall we calculate L ? We look at two adjacent deflections along the tube: $w(x)$ and $w(x+dx)$. The length of the tube's segment from x to $x+dx$, is given by $dl = \sqrt{dx^2 + dw^2} = dx \sqrt{1 + \left(\frac{dw}{dx}\right)^2}$. Since the tube length is much bigger than its maximal deflection ($w_{max}/L_0 \sim 50\text{nm}/1000\text{nm} = 1/20$), we can expand the square root and approximate dl : $dl = dx \sqrt{1 + \left(\frac{dw}{dx}\right)^2} \approx dx \left(1 + \frac{1}{2} \left(\frac{dw}{dx}\right)^2\right)$. If, for example, we assume that $w(x)$ is the first buckling solution of the EB equation $w(x) = \frac{1}{40} (1 - \cos(2\pi x/L_0))$ which satisfies the requirement that $\max(w(x))/L_0 = w(x=L_0/2)/L_0 = 1/20$, we can calculate the error in this approximation –

The total length of the deformed tube is given by:

$$L = \int_0^{L_0} dx \sqrt{1 + \left(\frac{dw}{dx}\right)^2} = \int_0^{L_0} dx \sqrt{1 + \left(\frac{2\pi}{40} \sin(2\pi x / L_0)\right)^2} \quad (2)$$

The approximate expression (L_{app}):

$$L_{app} = \int_0^{L_0} dx \left(1 + \frac{1}{2} \left(\frac{dw}{dx}\right)^2\right) = \int_0^{L_0} dx \left(1 + \frac{1}{2} \left(\frac{2\pi}{40} \sin(2\pi x / L_0)\right)^2\right) \quad (3)$$

The relative error between the two lengths is $\frac{|L-L_{app}|}{L} = 3 \cdot 10^{-5}$, which verify the validity of this approximation for our devices. If we return to the definition of the induced tension, we find:

$$T_{ind} = EA \frac{L-L_0}{L_0} = EA \left(\frac{1}{L_0} \int_0^{L_0} dx \sqrt{1 + \left(\frac{dw}{dx}\right)^2} - 1 \right) \approx \frac{EA}{2L_0} \int_0^{L_0} dx \left(\frac{dw}{dx}\right)^2 \quad (4)$$

After inserting this expression into Eq. 1 we find the source of the geometric non-linearity. The term $T_{ind}w''$ is an integro-differential non-linear term but the differential terms which are not within the integral are all linear. Hence, the linearity of the differential equation is preserved, and using the superposition principle is valid. To be more specific, imagine that $w(x) = w_1(x) + w_2(x)$ so what is legitimate is the following:

$$\begin{aligned} EI \frac{d^4}{dx^4} w_1(x) + T_0 \frac{d^2}{dx^2} w_1(x) - T_{ind}(w_1 + w_2) \frac{d^2}{dx^2} w_1(x) &= F_{ext} \\ EI \frac{d^4}{dx^4} w_2(x) + T_0 \frac{d^2}{dx^2} w_2(x) - T_{ind}(w_1 + w_2) \frac{d^2}{dx^2} w_2(x) &= F_{ext} \\ T_{ind}(w_1 + w_2) &= \frac{EA}{2L_0} \int_0^{L_0} dx \left(\frac{d(w_1 + w_2)}{dx}\right)^2 = \frac{EA}{2L_0} \int_0^{L_0} dx \left(\frac{dw_1}{dx} + \frac{dw_2}{dx}\right)^2 \end{aligned}$$

And the superposition principle of the linear terms allows us to write the following:

$$EI \frac{d^4}{dx^4} (w_1(x) + w_2(x)) + T_0 \frac{d^2}{dx^2} (w_1(x) + w_2(x)) - T_{ind}(w_1 + w_2) \frac{d^2}{dx^2} (w_1(x) + w_2(x)) = F_{ext}$$

Now, let's see what happens if the motion is not restricted to the x-z plane. Figure 2 presents two segments of a circle. The z axis marked as \mathbf{w} (in-plane deflection), and the y axis marked as \mathbf{v} (out of plane deflection). The longitudinal direction remains \mathbf{x} . The red segment presents the situation in which the deflection is solely within the x-z (or x-w) plane (red arrow). Then, we rotate the red segment by 45° and end up with the blue segment. At this specific configuration, if we wish to describe the segment

(or tube) deformation, we can use the red deformation and rotate it by $\theta=45^\circ$. Now, the description of the tube deformation consists of $w(x)$ (x-axis) as well as $v(x)$ (y-axis). Mathematically, we can express the relation between the red and blue deformations as the following:

$$\begin{aligned} w(x) &= w(x) \cdot \cos(\theta) \\ v(x) &= -w(x) \cdot \sin(\theta) \end{aligned} \quad (5)$$

Figure 2. Two circular segments with a relative $\theta=45^\circ$ rotation with respect to each other. left – 3D plot, right – projection on to the w - v (z - y) plane at $x=0$ (segment center).

Let's calculate the induced tension of the blue segment, and let's assume that the angle of rotation is general (θ):

$$\begin{aligned} T_{ind} &= \frac{EA}{2L_0} \int_0^{L_0} dx \left\{ \left(\frac{dw}{dx} \right)^2 + \left(\frac{dv}{dx} \right)^2 \right\} = \frac{EA}{2L_0} \int_0^{L_0} dx \left\{ \left(\frac{dw}{dx} \cos(\theta) \right)^2 + \left(\frac{dw}{dx} \sin(\theta) \right)^2 \right\} = \\ &= \frac{EA}{2L_0} \int_0^{L_0} dx \left(\frac{dw}{dx} \right)^2 (\cos^2(\theta) + \sin^2(\theta)) = \frac{EA}{2L_0} \int_0^{L_0} dx \left(\frac{dw}{dx} \right)^2 = T_{ind} \end{aligned} \quad (6)$$

As expected, since the circular shape remains the same, the induced tension should be identical. One can think about it as rotation of the reference coordinate system by $-\theta$ (minus θ). In other words, we could describe the circular deflection with vectorial deflection:

$$\vec{\psi}(x) = w(x)\hat{z} + v(x)\hat{y} \quad (7)$$

and the EB equation can be written as follows:

$$EI \frac{d^4}{dx^4} \vec{\psi}(x) + T_0 \frac{d^2}{dx^2} \vec{\psi}(x) - T_{ind} \frac{d^2}{dx^2} \vec{\psi}(x) = \vec{F}_{ext} \quad (8)$$

where \vec{F}_{ext} is the external force in the z and y direction (in our case the external force is solely in the z direction), and the induced tension is given by:

$$T_{ind} = \frac{EA}{2L_0} \int_0^{L_0} dx \frac{d\bar{\psi}}{dx} \cdot \frac{d\bar{\psi}}{dx} = \frac{EA}{2L_0} \int_0^{L_0} dx \left\{ \left(\frac{dw}{dx} \right)^2 + \left(\frac{dv}{dx} \right)^2 \right\} \quad (9)$$

If we decompose Eq. 8 to its components along the z and y directions, we find:

$$\hat{z}: \quad EI \frac{d^4}{dx^4} w(x) + T_0 \frac{d^2}{dx^2} w(x) - T_{ind} \frac{d^2}{dx^2} w(x) = F_{ext}^z \quad (10)$$

$$\hat{y}: \quad EI \frac{d^4}{dx^4} v(x) + T_0 \frac{d^2}{dx^2} v(x) - T_{ind} \frac{d^2}{dx^2} v(x) = F_{ext}^y \quad (11)$$

Let us conclude -

The induced tension is an important term. It is responsible for both the non-linearity of the EB equation, and for the coupling between the in-plane and out-of-plane motions.

This non-linear term is the origin for the Duffing oscillator in nano-beams, and responsible for numerous non-linear effects, such as parametric amplification, modes-coupling, and of-course the EB buckling instability and the snap-through transition. In our analysis we imply the superposition principle only for the linear terms, where for the non-linear terms we use the full expression without omitting any term.

This analysis which we describe is well known in the MEMS and NEMS community, and it is the foundation of any analysis in the field. In our manuscript, we followed the same analysis without any unjustified shortcut.

The second source of non-linearity arises from the external electric force of the local gate, which also depends on the tube deflection along the z-direction. This non-linearity was taken as well, and no superposition principle was implied to this term.

We have shown that for the non-linear term we use the full expression for the induced tension, whereas for the linear terms, we can imply the superposition principle without any fear. This formalism and technique are well justified, known and common in our field. We hope you find the above justification clearer and that, after reading this explanation, you are no longer concerned with misuse of the superposition principle.

Thanks to your comment, we have added the above clarification to the supplementary information under the new section titled “Linearization justification”.

We would like to thank all the reviewers again for their feedback, and for helping us to improve our manuscript.

Sincerely,

Mrs. Sharon Rechnitz
Dr. Tal Tabachnik
Dr. Michael Shlafman
Dr. Shlomo Shlafman
Prof. Yuval Yaish

REVIEWERS' COMMENTS

Reviewer #2 (Remarks to the Author):

I was unable to follow the explanation by the authors, in particular the 4 eqs (i-iv) reported after eq 4. The superposition principle implies that if w_1 and w_2 are solutions (of eq 1) related to F_{ext1} and F_{ext2} respectively also $w=w_1+w_2$ is solution (of eq 1) for $F_{ext}=F_{ext1}+F_{ext2}$, this implies the validity of eq iv, eq iii is valid by definition, but eqs i and ii are wrong, since they should present $Tind(w_1)$ and $Tind(w_2)$ respectively -in addition to F_{ext1} and F_{ext2} (eqs i-iv are even not self-consistent as can be noted summing i+ii that does not give iv since a factor of 2 is missing)- and not $Tind(w_1+w_2)$, thus basically the authors assume $Tind(w_1+w_2)=Tind(w_1)+Tind(w_2)$ (in addition to the mentioned problem on F_{ext}) that is wrong since $Tind(w)$ is nonlinear. It is impossible to demonstrate the general validity of the superposition principle in a nonlinear system by definition, for this reason I asked the evaluation of the error, not reported in the authors' answer (the estimated error concerning the Taylor expansion has nothing to do with superposition and is obvious, on this I agree).

31/08/2022

Dear Editor,

We would like to thank you for inviting a revision to our manuscript. Following is our detailed response to Reviewer #2's concern.

We hope you find this explanation sufficient and approve our manuscript for publication.

Reviewer #2

I was unable to follow the explanation by the authors, in particular the 4 eqs (i-iv) reported after eq 4. The superposition principle implies that if w_1 and w_2 are solutions (of eq 1) related to F_{ext1} and F_{ext2} respectively also $w=w_1+w_2$ is solution (of eq 1) for $F_{ext}=F_{ext1}+F_{ext2}$, this implies the validity of eq iv, eq iii is valid by definition, but eqs i and ii are wrong, since they should present $T_{ind}(w_1)$ and $T_{ind}(w_2)$ respectively -in addition to F_{ext1} and F_{ext2} (eqs i-iv are even not self-consistent as can be noted summing i+ii that does not give iv since a factor of 2 is missing)- and not $T_{ind}(w_1+w_2)$, thus basically the authors assume $T_{ind}(w_1+w_2)=T_{ind}(w_1)+T_{ind}(w_2)$ (in addition to the mentioned problem on F_{ext}) that is wrong since $T_{ind}(w)$ is nonlinear.

It is impossible to demonstrate the general validity of the superposition principle in a nonlinear system by definition, for this reason I asked the evaluation of the error, not reported in the authors' answer (the estimated error concerning the Taylor expansion has nothing to do with superposition and is obvious, on this I agree).

After carefully reading Reviewer #2 remarks, we decide to address them a little bit differently than before. We hope that this time our explanation will be clearer and satisfactory.

This time, our starting point is the complete Euler-Bernoulli (EB) beam equation. In the beginning, let's assume that the beam motion is restricted to the x-z plane (in-plane motion, $w(x,t)$, see Fig. 1).

$$EI \frac{\partial^4}{\partial x^4} w(x,t) + T_0 \frac{\partial^2}{\partial x^2} w(x,t) - T_{ind} \frac{\partial^2}{\partial x^2} w(x,t) + \rho A \frac{\partial^2}{\partial t^2} w(x,t) = F_{ext}(w(x,t)) \quad (1)$$

where EI is the bending rigidity, T_0 is the initial tension, ρ is the mass density, A is the CNT cross section, and T_{ind} is the induced tension resulted from the CNT movements (both static as well as dynamic motions) and is given by the following expression:

$$T_{ind} = EA \frac{L-L_0}{L_0} = EA \left(\frac{1}{L_0} \int_0^{L_0} dx \sqrt{1 + \left(\frac{dw}{dx} \right)^2} - 1 \right) \approx \frac{EA}{2L_0} \int_0^{L_0} dx \left(\frac{dw}{dx} \right)^2 \quad (2)$$

Figure 1. Schematic of a typical bi-stable CNT device with initial upward buckling and our coordinates system.

Regarding the external force per unit length exerted by the local gate - the expression for the force is given by:

$$F_{ext} = \frac{1}{2} \frac{\partial C_{gate}(z)}{\partial z} V_g^2 = \frac{1}{g_0 - W} \frac{\pi \epsilon_0}{\ln(2(g_0 - W)/r)} (V_{DC} + V_{AC})^2 \approx \frac{1}{g_0 - W} \frac{\pi \epsilon_0}{\ln(2g_0/r)} (V_{DC} + V_{AC})^2 \quad (3)$$

where g_0 is the distance between the electrodes and the local gate, r is the tube radius, and V_{DC}/V_{AC} is the DC/AC component of the gate voltage. The approximation we use ignores W within the logarithmic function compared to g_0 . We estimate this approximation using the complete solution and found it to be less than 5%. Since in all our measurements $V_{DC} \gg V_{AC}$ we can write the external force as follows:

$$F_{ext} = \frac{1}{g_0 - W} \frac{\pi \epsilon_0}{\ln(2g_0/r)} (V_{DC} + 2 \cdot V_{DC} \cdot V_{AC}) = F_{ext}^{DC} + F_{ext}^{AC} \quad (4)$$

In the beginning of this reply, in order to facilitate our explanation, we assume that F_{ele} is not dependent on the distance between the CNT and the local gate. Afterwards, we will present the situation where the external force also depends on distance. Thus, the EB equation reads as follows:

$$EI \frac{\partial^4}{\partial x^4} w(x,t) + T_0 \frac{\partial^2}{\partial x^2} w(x,t) - \frac{\partial^2}{\partial x^2} w(x,t) \frac{EA}{2L_0} \int_0^{L_0} dx \left(\frac{\partial w}{\partial x} \right)^2 + \rho A \frac{\partial^2}{\partial t^2} w(x,t) = F_{ext} = F_{ext}^{DC} + F_{ext}^{AC} \quad (5)$$

The complete CNT in plane displacement can be written as follows:

$$w(x,t) = w_s(x) + w_d(x,t) \quad (6)$$

where $W_s(x)/W_d(x,t)$ is the static/dynamic response to the external force. After inserting this relation into Eq. 5 we obtain:

$$EI \frac{\partial^4}{\partial x^4} (w_s(x) + w_d(x,t)) + T_0 \frac{\partial^2}{\partial x^2} (w_s(x) + w_d(x,t)) - \frac{\partial^2}{\partial x^2} (w_s(x) + w_d(x,t)) \cdot \frac{EA}{2L_0} \int_0^{L_0} dx \left(\frac{\partial w_s}{\partial x} + \frac{\partial w_d}{\partial x} \right)^2 + \rho A \frac{\partial^2}{\partial t^2} w_d(x,t) = F_{ext}^{DC} + F_{ext}^{AC} \quad (7)$$

and after opening the square term in the induced tension we receive:

$$EI \frac{\partial^4}{\partial x^4} (w_s(x) + w_d(x,t)) + T_0 \frac{\partial^2}{\partial x^2} (w_s(x) + w_d(x,t)) - \frac{\partial^2}{\partial x^2} (w_s(x) + w_d(x,t)) \cdot \frac{EA}{2L_0} \int_0^{L_0} dx \left(\left(\frac{\partial w_s}{\partial x} \right)^2 + \left(\frac{\partial w_d}{\partial x} \right)^2 + 2 \frac{\partial w_s}{\partial x} \frac{\partial w_d}{\partial x} \right) + \rho A \frac{\partial^2}{\partial t^2} w_d(x,t) = F_{ext}^{DC} + F_{ext}^{AC} \quad (8)$$

Now, it is possible to split this equation into static and dynamic parts:

$$(a) \quad EI \frac{\partial^4}{\partial x^4} w_s(x) + T_0 \frac{\partial^2}{\partial x^2} w_s(x) - \frac{\partial^2}{\partial x^2} w_s(x) \cdot \frac{EA}{2L_0} \int_0^{L_0} dx \left(\frac{\partial w_s}{\partial x} \right)^2 = F_{ext}^{DC}$$

$$(b) \quad EI \frac{\partial^4}{\partial x^4} w_d(x,t) + T_0 \frac{\partial^2}{\partial x^2} w_d(x,t) - \frac{\partial^2}{\partial x^2} w_s(x) \cdot \frac{EA}{2L_0} \int_0^{L_0} dx \left(\left(\frac{\partial w_d}{\partial x} \right)^2 + 2 \frac{\partial w_s}{\partial x} \frac{\partial w_d}{\partial x} \right) - \frac{\partial^2}{\partial x^2} w_d(x,t) \cdot \frac{EA}{2L_0} \int_0^{L_0} dx \left(\left(\frac{\partial w_s}{\partial x} \right)^2 + \left(\frac{\partial w_d}{\partial x} \right)^2 + 2 \frac{\partial w_s}{\partial x} \frac{\partial w_d}{\partial x} \right) + \rho A \frac{\partial^2}{\partial t^2} w_d(x,t) = F_{ext}^{AC} \quad (9)$$

This example presents how we split the full displacement $W(x,t)$ to its static and dynamic parts. All the terms beside the induced tension are linear, thus this operation was possible. The non-linear term indeed contains cross terms and quadratic and cubic terms and all of them were taken into account in our calculation. For example, if one is interested in the response to the DC force, he should solve Eq. 9a. If one is interested in the vibrational modes, he assumes that $W_d \ll W_s$, and he may ignore quadratic and cubic terms of W_d (the AC force can be small as one desires, making the dynamic response small as needed). However, cross terms of W_s and W_d must be considered. Specifically, these terms are the origin for the unique dependence of the vibrational modes for arch shaped beams, or CNTs. After linearization with respect to W_d in order to find the vibrational modes, Eq. 9b is written as follows:

$$(b) \quad EI \frac{\partial^4}{\partial x^4} w_d(x,t) + T_0 \frac{\partial^2}{\partial x^2} w_d(x,t) - \frac{\partial^2}{\partial x^2} w_s(x) \cdot \frac{EA}{2L_0} \int_0^{L_0} 2 \frac{\partial w_s}{\partial x} \frac{\partial w_d}{\partial x} dx \quad (10)$$

$$- \frac{\partial^2}{\partial x^2} w_d(x,t) \cdot \frac{EA}{2L_0} \int_0^{L_0} dx \left(\frac{\partial w_s}{\partial x} \right)^2 + \rho A \frac{\partial^2}{\partial t^2} w_d(x,t) = 0$$

The AC force is set to zero in order to find the resonance vibrational modes.

Now we can move to the case where also the out-of-plane motion is possible, but the external force is still along the z direction. Instead of single EB equation, we have two, one for the z direction and one for the y direction. The CNT displacement can be described in a vectorial form:

$$\vec{\psi}(x,t) = w(x,t)\vec{z} + v(x,t)\vec{y} \quad (11)$$

and the EB equation translates into:

$$EI \frac{\partial^4}{\partial x^4} \vec{\psi}(x,t) + T_0 \frac{\partial^2}{\partial x^2} \vec{\psi}(x,t) - \frac{\partial^2}{\partial x^2} \vec{\psi}(x,t) \frac{EA}{2L_0} \int_0^{L_0} dx \left(\frac{\partial \vec{\psi}}{\partial x} \cdot \frac{\partial \vec{\psi}}{\partial x} \right) + \rho A \frac{\partial^2}{\partial t^2} \vec{\psi}(x,t) = \vec{F}_{ext} = F_{ext}^{DC} \vec{z} + F_{ext}^{AC} \vec{z} \quad (12)$$

The dot inside the integral presents a scalar product between the two vectors. Eq. 12 can be split into two EB equations for each direction, i.e.:

$$\begin{aligned} \hat{z}: \quad & EI \frac{\partial^4}{\partial x^4} w(x,t) + T_0 \frac{\partial^2}{\partial x^2} w(x,t) - \frac{\partial^2}{\partial x^2} w(x,t) \frac{EA}{2L_0} \int_0^{L_0} dx \left(\frac{\partial w}{\partial x} \frac{\partial w}{\partial x} + \frac{\partial v}{\partial x} \frac{\partial v}{\partial x} \right) + \rho A \frac{\partial^2}{\partial t^2} w(x,t) = \\ & F_{ext}^{DC} + F_{ext}^{AC} \\ \hat{y}: \quad & EI \frac{\partial^4}{\partial x^4} v(x,t) + T_0 \frac{\partial^2}{\partial x^2} v(x,t) - \frac{\partial^2}{\partial x^2} v(x,t) \frac{EA}{2L_0} \int_0^{L_0} dx \left(\frac{\partial w}{\partial x} \frac{\partial w}{\partial x} + \frac{\partial v}{\partial x} \frac{\partial v}{\partial x} \right) + \rho A \frac{\partial^2}{\partial t^2} v(x,t) = 0 \end{aligned} \quad (13)$$

One can notice the importance of the induced tension term. This term is responsible for the coupling between the in-plane and out-of-plane motions. Now, we can proceed as before in Eq. 6. We express $W(x,t)$ and $V(x,t)$ as sums of static and dynamic parts, as follows:

$$\begin{aligned} w(x,t) &= w_s(x) + w_d(x,t) \\ v(x,t) &= v_s(x) + v_d(x,t) \end{aligned} \quad (14)$$

And we can plug these expressions into Eqs. 13. The results are the following after splitting these equations into static and dynamic response:

$$\begin{aligned}
\hat{z}(static): \quad & EI \frac{\partial^4}{\partial x^4} w_s(x) + T_0 \frac{\partial^2}{\partial x^2} w_s(x) - \frac{\partial^2}{\partial x^2} w_s(x) \frac{EA}{2L_0} \int_0^{L_0} dx \left(\frac{\partial w_s}{\partial x} \frac{\partial w_s}{\partial x} + \frac{\partial v_s}{\partial x} \frac{\partial v_s}{\partial x} \right) = F_{ext}^{DC} \\
\hat{z}(dynamic): \quad & EI \frac{\partial^4}{\partial x^4} w_d(x,t) + T_0 \frac{\partial^2}{\partial x^2} w_d(x,t) - \\
& \frac{\partial^2}{\partial x^2} w_d(x) \cdot \frac{EA}{2L_0} \int_0^{L_0} dx \left(\frac{\partial w_d}{\partial x} \left(2 \frac{\partial w_s}{\partial x} + \frac{\partial w_d}{\partial x} \right) + \frac{\partial v_d}{\partial x} \left(2 \frac{\partial v_s}{\partial x} + \frac{\partial v_d}{\partial x} \right) + \frac{\partial w_s}{\partial x} \frac{\partial w_s}{\partial x} + \frac{\partial v_s}{\partial x} \frac{\partial v_s}{\partial x} \right) - \\
& \frac{\partial^2}{\partial x^2} w_s(x) \cdot \frac{EA}{2L_0} \int_0^{L_0} dx \left(\frac{\partial w_d}{\partial x} \left(2 \frac{\partial w_s}{\partial x} + \frac{\partial w_d}{\partial x} \right) + \frac{\partial v_d}{\partial x} \left(2 \frac{\partial v_s}{\partial x} + \frac{\partial v_d}{\partial x} \right) \right) + \rho A \frac{\partial^2}{\partial t^2} w_d(x,t) = F_{ext}^{AC} \\
\hat{y}(static): \quad & EI \frac{\partial^4}{\partial x^4} v_s(x) + T_0 \frac{\partial^2}{\partial x^2} v_s(x) - \frac{\partial^2}{\partial x^2} v_s(x) \frac{EA}{2L_0} \int_0^{L_0} dx \left(\frac{\partial w_s}{\partial x} \frac{\partial w_s}{\partial x} + \frac{\partial v_s}{\partial x} \frac{\partial v_s}{\partial x} \right) = 0 \\
\hat{y}(dynamic): \quad & EI \frac{\partial^4}{\partial x^4} v_d(x,t) + T_0 \frac{\partial^2}{\partial x^2} v_d(x,t) - \\
& \frac{\partial^2}{\partial x^2} v_d(x) \cdot \frac{EA}{2L_0} \int_0^{L_0} dx \left(\frac{\partial w_d}{\partial x} \left(2 \frac{\partial w_s}{\partial x} + \frac{\partial w_d}{\partial x} \right) + \frac{\partial v_d}{\partial x} \left(2 \frac{\partial v_s}{\partial x} + \frac{\partial v_d}{\partial x} \right) + \frac{\partial w_s}{\partial x} \frac{\partial w_s}{\partial x} + \frac{\partial v_s}{\partial x} \frac{\partial v_s}{\partial x} \right) - \\
& \frac{\partial^2}{\partial x^2} v_s(x) \cdot \frac{EA}{2L_0} \int_0^{L_0} dx \left(\frac{\partial w_d}{\partial x} \left(2 \frac{\partial w_s}{\partial x} + \frac{\partial w_d}{\partial x} \right) + \frac{\partial v_d}{\partial x} \left(2 \frac{\partial v_s}{\partial x} + \frac{\partial v_d}{\partial x} \right) \right) + \rho A \frac{\partial^2}{\partial t^2} v_d(x,t) = 0
\end{aligned} \tag{15}$$

As one can see, all the non-linear terms are included within these equations. We did not neglect any of them. The linear terms indeed follow the super position principle, but we agree that this notation can be misleading. In order to find the vibrational modes, we first solve the coupled static equations (Eq. 15, for W_s and V_s) and after we linearize the dynamic parts of Eq. 15 with respect to W_d and V_d , i.e.:

$$\begin{aligned}
\hat{z}(dynamic): \quad & EI \frac{\partial^4}{\partial x^4} w_d(x,t) + T_0 \frac{\partial^2}{\partial x^2} w_d(x,t) - \frac{\partial^2}{\partial x^2} w_d(x) \cdot \frac{EA}{2L_0} \int_0^{L_0} dx \left(\frac{\partial w_s}{\partial x} \frac{\partial w_s}{\partial x} + \frac{\partial v_s}{\partial x} \frac{\partial v_s}{\partial x} \right) - \\
& \frac{\partial^2}{\partial x^2} w_s(x) \cdot \frac{EA}{2L_0} \int_0^{L_0} dx \left(2 \frac{\partial w_d}{\partial x} \frac{\partial w_s}{\partial x} + 2 \frac{\partial v_d}{\partial x} \frac{\partial v_s}{\partial x} \right) + \rho A \frac{\partial^2}{\partial t^2} w_d(x,t) = 0 \\
\hat{y}(dynamic): \quad & EI \frac{\partial^4}{\partial x^4} v_d(x,t) + T_0 \frac{\partial^2}{\partial x^2} v_d(x,t) - \frac{\partial^2}{\partial x^2} v_d(x) \cdot \frac{EA}{2L_0} \int_0^{L_0} dx \left(\frac{\partial w_s}{\partial x} \frac{\partial w_s}{\partial x} + \frac{\partial v_s}{\partial x} \frac{\partial v_s}{\partial x} \right) - \\
& \frac{\partial^2}{\partial x^2} v_s(x) \cdot \frac{EA}{2L_0} \int_0^{L_0} dx \left(2 \frac{\partial w_d}{\partial x} \frac{\partial w_s}{\partial x} + 2 \frac{\partial v_d}{\partial x} \frac{\partial v_s}{\partial x} \right) + \rho A \frac{\partial^2}{\partial t^2} v_d(x,t) = 0
\end{aligned} \tag{16}$$

These two equations can be solved simultaneously, and the in-plane and out-plane vibrational modes are found.

Now, we can proceed to the more realistic situation where the external electric force exerted by the local gate indeed depends also on the CNT vertical position ($W(x,t)$). In order not to be too lengthy, we will address only the case where in-plane motion is allowed, however, the same arguments and procedure hold also for the general case where in-plane and out-of-plane motions are possible. The general EB equation reads as follows:

$$\begin{aligned}
& EI \frac{\partial^4}{\partial x^4} (w_s(x) + w_d(x,t)) + T_0 \frac{\partial^2}{\partial x^2} (w_s(x) + w_d(x,t)) - \frac{\partial^2}{\partial x^2} (w_s(x) + w_d(x,t)) \cdot \\
& \frac{EA}{2L_0} \int_0^{L_0} dx \left(\left(\frac{\partial w_s}{\partial x} \right)^2 + \left(\frac{\partial w_d}{\partial x} \right)^2 + 2 \frac{\partial w_s}{\partial x} \frac{\partial w_d}{\partial x} \right) + \rho A \frac{\partial^2}{\partial t^2} w_d(x,t) = \\
& \frac{1}{g_0 - (w_s(x) + w_d(x,t))} \frac{\pi \epsilon_0}{\ln(2g_0/r)} (V_{DC} + 2 \cdot V_{DC} \cdot V_{AC})
\end{aligned} \tag{17}$$

Now, we multiply Eq. 17 by $g_0 - (w_s(x) + w_d(x,t))$ to obtain the following:

$$\begin{aligned}
& (g_0 - (w_s(x) + w_d(x,t))) \cdot EI \frac{\partial^4}{\partial x^4} (w_s(x) + w_d(x,t)) + (g_0 - (w_s(x) + w_d(x,t))) \cdot T_0 \frac{\partial^2}{\partial x^2} (w_s(x) + w_d(x,t)) - \\
& (g_0 - (w_s(x) + w_d(x,t))) \cdot \frac{\partial^2}{\partial x^2} (w_s(x) + w_d(x,t)) \cdot \frac{EA}{2L_0} \int_0^{L_0} dx \left(\left(\frac{\partial w_s}{\partial x} \right)^2 + \left(\frac{\partial w_d}{\partial x} \right)^2 + 2 \frac{\partial w_s}{\partial x} \frac{\partial w_d}{\partial x} \right) + \\
& (g_0 - (w_s(x) + w_d(x,t))) \cdot \rho A \frac{\partial^2}{\partial t^2} w_d(x,t) = \frac{\pi \epsilon_0}{\ln(2g_0/r)} (V_{DC} + 2 \cdot V_{DC} \cdot V_{AC})
\end{aligned} \tag{18}$$

Now, we follow the same procedure as before, meaning, split Eq. 18 to static and dynamic equations, keeping all the linear and non-linear terms. As one can see, here, the non-linear terms arise both from the induced tension term, but also from the external force (through the multiplication of $g_0 - (w_s(x) + w_d(x,t))$). The analysis within the manuscript is identical to what we describe here.

We do hope that this document assist explaining our procedure and analysis.

Sincerely,

Sharon Rechnitz, Tal Tabachnik, Michael Shlafman, Shlomo Shlafman, and Yuval Yaish